# Carbonate Soil Cryogenesis in South Yakutia (Russia)

**Andrey Melnikov** [1,*] , **Anna Kut** [1], **Ze Zhang** [2]  **and Viktor Rochev** [3]

1. Melnikov Permafrost Institute, Siberian Branch, Russian Academy of Science, 677010 Yakutsk, Russia; ann.urban@mail.ru
2. Institute of Cold Regions Science and Engineering, School of Civil Engineering, Northeast Forestry University, Harbin 150040, China; zez@nefu.edu.cn
3. Technical Institute (Branch), M. K. Ammosov North-Eastern Federal University, 678960 Neryungri, Russia; viktor-rochev74@mail.ru
* Correspondence: MelnikowDron@mail.ru

**Abstract:** The present study investigates changes occurring in the material composition and properties of the South Yakutian carbonate soils during cryogenesis. The nature of the transformations of certain limestone varieties composing the surfaces of rock massifs was determined using scanning electron microscopy, 3D X-ray tomography, as well as lithological–mineralogical and optical–petrographic studies, over a 10-year period. The areas in carbonate rock massifs with increased clay content, pyritisation, dolomitisation, and baritisation, as well as zones of calcite and dolomite junction, were found to be least resistant to the effects of processes associated with water phase transitions, i.e., freezing and thawing. The mineral proportion of limestone on the surface of soil massifs chemically processed over a 10-year period reached 5–7% of the volume of the weathered rocks. In the process of transformation, not only the composition of the rocks changed, but also the nature of the structural bonds that significantly influence their mechanical strength properties. The number of cracks for weathered soil samples increased by 9–16%; their opening increased by 13–18%. For rocks initially having uniaxial compression strength in the range of 33–46 MPa, this strength was reduced by 19–27%. Laboratory experiments on 1000-fold cyclic freezing and thawing of carbonate rock samples (which corresponds to an 8–10-year period of weathering on the surface of a mountain outcrop under the natural conditions of South Yakutia) demonstrate the similarity of these changes with those observed in samples taken from the sides of open pits 10 years ago. In general, soils are influenced by a wide range of environmental factors under natural conditions. The significant influence of alternating temperatures on the changes in the composition and structure of limestones in South Yakutia is characterised in detail.

**Keywords:** limestone; cryogenesis; matter alteration; South Yakutia

## 1. Introduction

Soils undergo various transformations under the influence of processes associated with freezing and thawing water phase transitions, resulting in changes in their properties [1–6]. The combination of these processes is referred to as cryogenesis [7]. To date, the main regularities of such changes have been established for most soil types having rigid structural bonds of crystallisation or cementation type (such soils defined as rocky according to the classification system adopted in the Russian Federation are characterised by a uniaxial compression strength of at least 5 MPa [8]) under laboratory conditions. As a rule, these studies are based on determining the physical and strength characteristics of soil samples after repeated cycles of freezing and thawing [1,9–17]. Very little work has been carried out to describe the mechanisms (physico-chemical, mineralogical, structural, including the nature of water migration and ice formation) and material transformations that occur during phase transitions of water, i.e., freezing and thawing. Nevertheless, the establishment of soil transformation features during cryogenesis at the micro-scale

is of particular value for explaining the specific engineering and geological properties of soils having structural relationships dependent on their cyclic freezing–thawing. For example, optical–petrographic studies were carried out on the host rocks of some coal deposits of Yakutia to explain the overestimation of strength characteristics of the rocks of individual quarry benches during mining operations. Analytical studies of samples from boreholes identified the development zones of ankeritisation processes, to which the local strengthening of the soil massif turned out to be confined [18,19].

Therefore, in order to determine the features of cryogenically driven alteration of the rocky soils of South Yakutia, as well as to augment the factual material base, a comprehensive laboratory study of certain carbonate rock varieties in the region was carried out. South Yakutia was chosen as a testing ground due to the lack of specific studies of soil cryogenesis in the region. The high density of potential factors determining the intensity of rock weathering [11,20,21] allows the results of laboratory work and field observations to be compared in the future.

## 2. Materials and Methods

The study area is located in the Western Yangi mountain range, which rises above the Aldan plateau and is composed of Archean gneisses and granites (Figure 1a,c). This massif comprises laccoliths of post-Jurassic rocks formed by erosion and denudation, represented by syenite–porphyry. The absolute elevation of the highest point in the area (Gora Evota) is 1603 m. The southern part of the study area comprising part of the Chulman plateau is composed of Cambrian carbonate, as well as terrigenous sedimentary sandstones, aleurolites and argillites of the Jurassic and Lower Cretaceous periods. Here the absolute elevations do not exceed 1300–1400 m.

The severe climate of the region is classified as sharply continental. According to data acquired from the meteorological station located 60 km from Gora Evota, the average annual air temperature is minus 10 °C, while the average monthly minima and maxima are minus 39 °C and plus 23 °C, respectively. Thus, the maximum amplitude of air temperature fluctuations exceeds 80 degrees, while their daily amplitude can reach 40 degrees. The number of temperature transitions through 0 °C per year on the surface of the studied soil massifs exceeds 100 cycles (according to the results of the authors' observations in 2018–2019). Although the thickness of the snow cover in the region does not exceed half a metre on average, it is distributed unevenly due to the activity of the wind. As a result, up to 3 m of snow can accumulate on leeward slopes. A stable snow cover persists from September to early June, with some snowfields disappearing only by the end of summer or even continuing into the following year.

The harsh climate affects the soil-forming process, which is characterised by the suppression of chemical and biochemical forms of weathering and, conversely, by the high intensity of physical weathering. For this reason, coarse–skeletal tundra–taiga podzolised soils are well represented in the study area.

Permafrost in the study area has both continuous and mostly continuous distribution according to the absolute elevation (Figure 1b). The boundary between these types of distribution roughly coincides with the upper border of the mountain taiga, i.e., at an absolute elevation of 1100–1200 m. At elevations of 1500 m, it reaches a thickness of 500 m at a rock temperature of –7 to –8 °C at a zero fluctuation depth. Further down the slopes, the thickness of the frozen strata decreases to several tens of metres. The thickness of seasonal thawing in the area ranges from 0.3 to 3 m. The effect of cryogenesis on carbonate massifs was assessed using soil samples taken from boreholes drilled in 2020. These samples were compared with those obtained from the walls of mines having a known opening date (at least 10 years ago) (Figure 2a,b). Borehole samples were correlated with the sections of corresponding walls of ditches, trenches and quarries. In other words, "fresh" rock samples obtained from boreholes were compared with "weathered" samples. The "fresh" samples were additionally subjected to multiple cycles of alternating freezing and thawing under laboratory conditions (1000 cycles) and compared with those taken from mine walls.

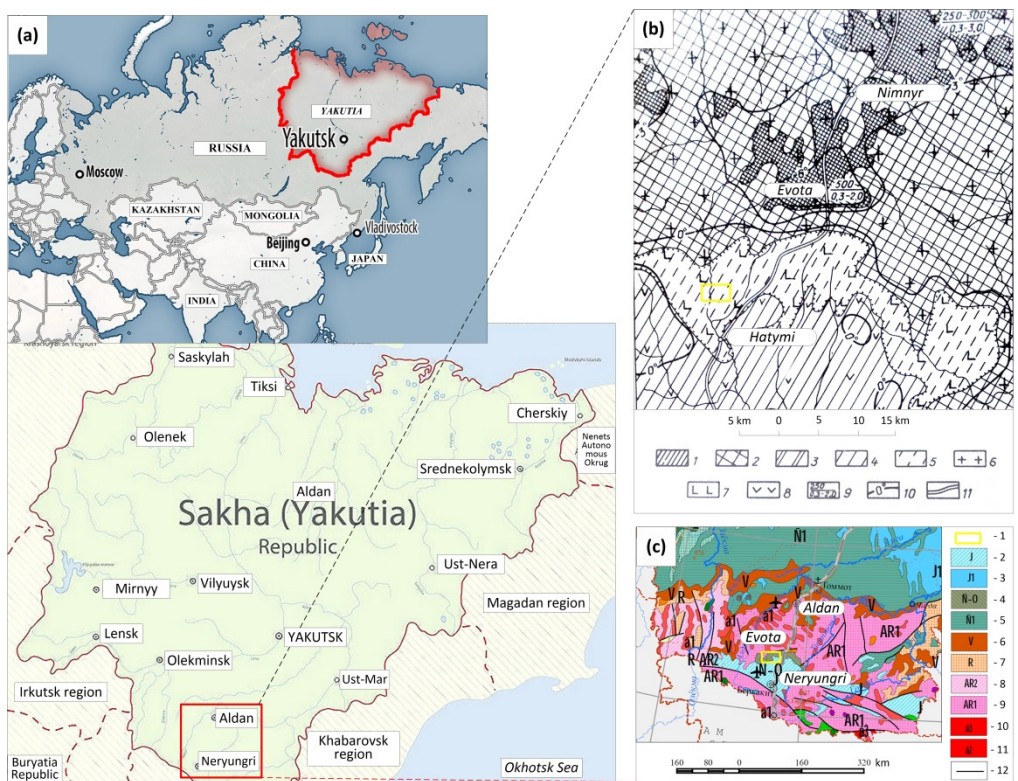

**Figure 1.** Overview map of the study area: (**a**) location of the study area; (**b**) geocryological scheme of the research area (according to V.R. Alekseev, 1963 [22]). The yellow outline marks the boundaries of the study area. Types of permafrost rocks by the degree of discontinuity: 1—continuous (100%); 2—mostly continuous (95–98%); 3—intermittent (70–80%); 4—intermittent (50%); 5—insular (25–30%). Types of permafrost in composition: 6—massive; 7—dense cemented Cambrian marine sediments; 8—dense cemented Jurassic continental sediments; 9—maximum permafrost thickness values, m (in the numerator) and typical seasonal thawing thickness limits, m (in the denominator); 10—isotherms at the bottom of the layer with annual zero-point oscillations (15–20 m for watersheds); 11—highway. (**c**) Fragment of the geological map of the study area (map prepared by the A.P. Karpinsky All-Russian Geological Research Institute within the framework of the State Assignment of the Federal Agency for Subsoil Use dated 26 December 2019 No. 049-00017-20-04): 1—boundaries of the study area; 2—Jurassic system, undivided; 3—Jurassic system, lower section; 4—Ordovician system, Cambrian–Ordovician; 5—Cambrian system, lower section; 6—wend; 7—Riphean; 8—upper archaea; 9—lower archaea; 10—felsic intrusive rocks, Mesozoic; 11—felsic intrusive rocks, Palaeozoic; 12—faults on dry land. For testing, cubic samples with dimensions of 4.0 cm × 4.0 cm × 4.0 cm were prepared from the selected material (Figure 2c); the size is regulated by the interstate standard of the Russian Federation.

Experiments on the cyclic freezing and thawing of samples under laboratory conditions included the selection of reference samples, as well as studies of their material composition and structure in predetermined and labelled areas. The samples were then subjected to alternate freezing and thawing. Following the completion of the test cycle, the same studies were carried out using the same sample areas.

Freezing conditions for samples was minus 20 °C in a freezer with the temperature maintained to an accuracy of up to 2 °C. The holding time of the samples in the freezer at which the temperature was stabilised in the cube-shaped sample was determined experimentally prior to the start of the experiments by installing thermistors. Thus, the freezing time of the samples was at least 8 h. The samples were thawed at a temperature of between +18 and +20 °C for at least 10 h. The soils were subjected to 1000 cycles of

alternate freezing and thawing. The method used for the cyclic freezing and thawing of soils is described in detail in [11,21].

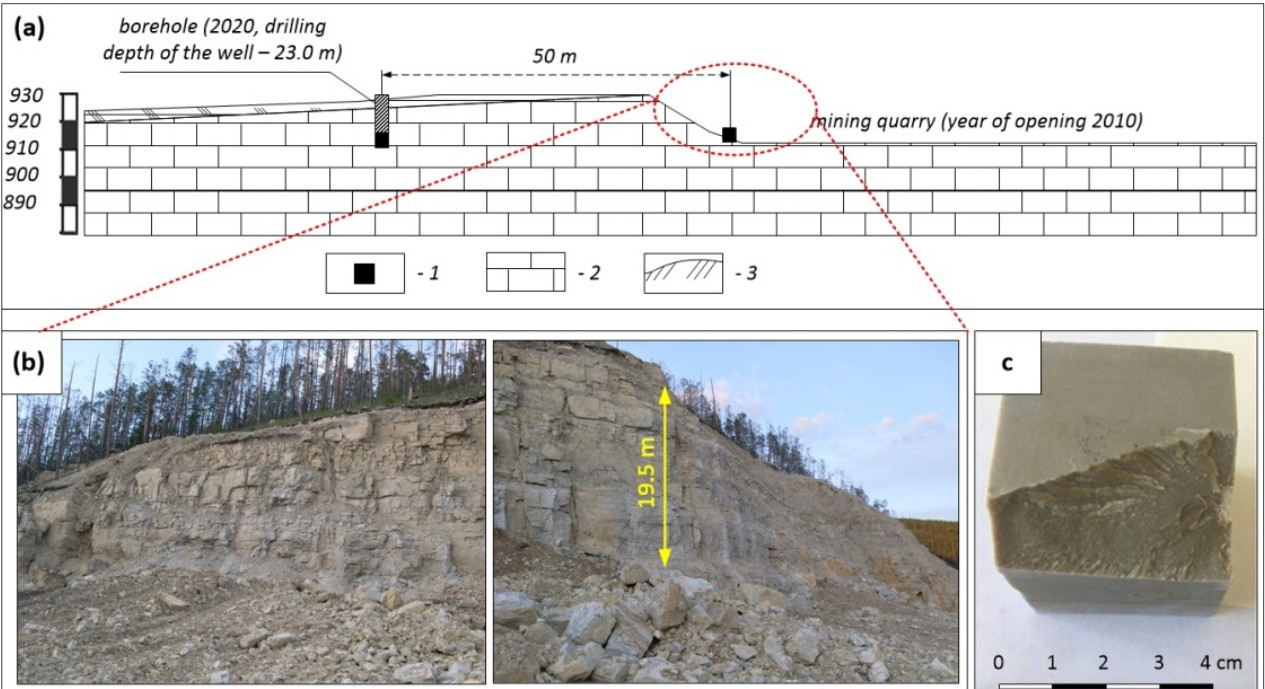

**Figure 2.** Scheme of the selection of limestone from the soil mass for analytical studies: (**a**) soil sampling scheme: 1—soil sampling points, 2—carbonate massif, 3—soil and vegetation layer; (**b**) general view of the wall of the mine from the base of which the soil sample was taken; (**c**) prepared cube-shaped specimen with cut-off top.

In general, analytical works included the following:

(1)    Lithological–mineralogical and optical–petrographic studies, which were carried out to obtain the data on soil composition and structural features. The studies were carried out using a Nikon SMZ 645 binocular microscope (Nikon, Tokyo, Japan) and an Olympus BX51 polarising microscope (Olympus Corporation, Tokyo, Japan);

(2)    Scanning electron microscopy (SEM) carried out on a JSM 6390LV microscope (JEOL, Japan) with a resolution of up to 3 nm and a maximum magnification of up to 300,000 times in order to study the surface microrelief of minerals composing the soil samples, as well as to assess micro-fracturing and microporosity;

(3)    3D X-ray tomography performed to obtain three-dimensional data on changes in structural and textural characteristics and porosity-fracturing of samples. The set of numerical data with fracture opening sizes obtained during the measurement was visualised in the form of a sample fracture map and a histogram of fracture size (opening) distribution (Figure 3);

(4)    Chemical analyses of minerals carried out using an INCA ENERGY 350 and INCA WAVE 500 energy- and wavelength-dispersive spectrometers ("OXFORD INSTRU-MENTS", Abingdon, UK);

(5)    Determination of physical and mechanical properties—density and uniaxial compressive strength.

The properties of the following soil varieties were analysed: dolomitic limestone, and organogenic limestone with dolomite.

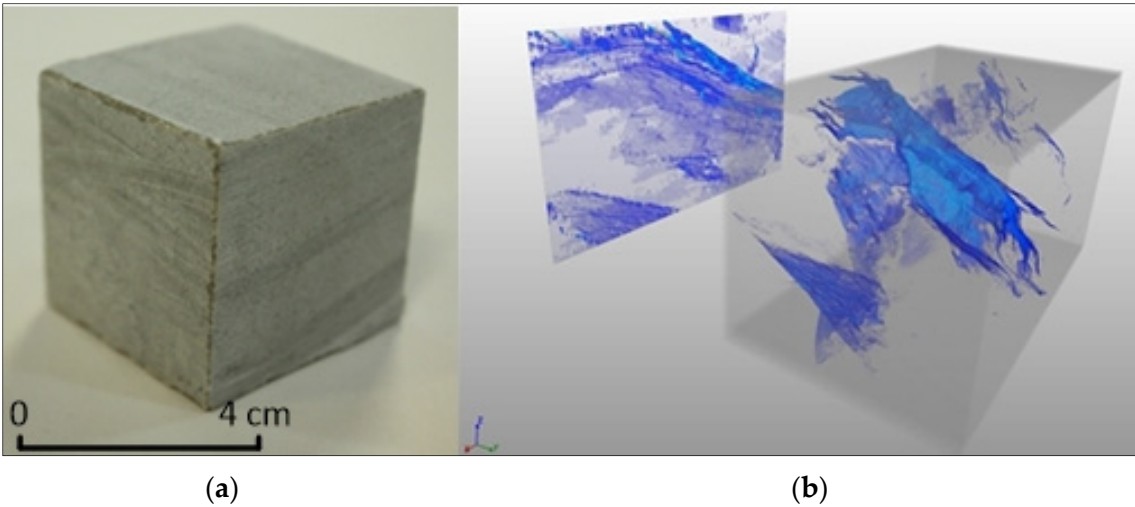

**Figure 3.** Construction of a soil sample fracture map using X-ray 3D tomography: (**a**) limestone sample prepared for X-ray 3D tomography; (**b**) a sample of the fracture map (**left**) and the void fracture space of the study area (**right**), obtained during the analysis of soil samples using a microfocus X-ray control system with a computed tomography function (cracks are shown in shades of blue).

## 3. Results

### 3.1. Dolomitic Limestone

The following features of the chemical composition of minerals (according to electron microprobe analysis data) are characteristic of the limestone samples obtained from boreholes (Table 1). Typically, three zones were distinguished in the studied soils, namely pure limestones, dolomitised limestones and silicified limestones. The former were also characterised by the presence of silica (up to 2.03%) and alumina (up to 0.60%), as well as a low magnesium (MgO up to 0.91%) and iron (up to 0.41%) content. In rare dolomitisation zones, iron oxides were present in small amounts (up to 0.90%). In silicification zones, the silica content noticeably increased (up to 10.91%) along with the appearance of elements such as Al, K, Na, S and Cl. In addition, the rock contained quartz veinlets and barite inclusions. Silicification zones are typically accompanied by baritisation zones reflecting the impact of metasomatic processes on the rock.

**Table 1.** Chemical composition of minerals (according to electron microprobe analyses) of dolomitic limestone samples taken from the borehole core, wt. %.

| Oxide | Calcite | Silicified Calcite | Calcite | Calcite |
|---|---|---|---|---|
| CaO | 51.85 | 50.25 | 53.63 | 51.74 |
| MgO | 0.71 | 1.04 | 0.91 | 1.00 |
| $SiO_2$ | 1.45 | 10.91 | 0.76 | 4.38 |
| $Al_2O_3$ | 0.40 | 1.16 | 0.43 | 0.51 |
| FeO | 0.41 | 0.00 | 0.17 | 0.66 |
| MnO | 0.00 | 0.00 | 0.00 | 0.00 |
| $K_2O$ | 0.19 | 0.71 | 0.00 | 0.00 |
| $Na_2O$ | 0.00 | 1.40 | 0.00 | 0.71 |
| $SO_3$ | 0.00 | 1.07 | 0.00 | 0.00 |
| Cl | 0.00 | 0.47 | 0.00 | 0.00 |

In terms of chemical composition, the limestones from outcrops were distinguished from the borehole samples by signs of weak phosphatisation (indicator elements—P, F), sulfatisation (S) and chloridisation (Cl) (Table 2). At the same time, areas with more intense phosphatisation, sulfidisation (pyrite and chalcopyrite) and leucoxenisation and, in some cases, with carbonaceous matter, were observed in the limestone samples from the outcrop surfaces located within the border of the river valley, i.e., in a more humid environment.

**Table 2.** Chemical composition of minerals (according to electron microprobe analyses) of dolomitic limestone samples taken from the surface of the mine walls, wt. %.

| Oxide | Leucoxenisation Zone | Leucoxenisation Zone | Calcite | Quartz | Calcite | Calcite | Ankerite | Ankerite | Quartz | Calcite |
|---|---|---|---|---|---|---|---|---|---|---|
| CaO | 3.34 | 2.66 | 52.74 | 0.90 | 54.16 | 54.92 | 33.40 | 35.18 | 0.73 | 53.92 |
| MgO | 0.26 | 0.27 | 0.16 | 0.00 | 0.54 | 0.47 | 12.90 | 14.91 | 0.27 | 0.63 |
| SiO2 | 4.94 | 3.83 | 1.41 | 98.87 | 0.44 | 0.21 | 0.46 | 0.00 | 97.92 | 0.58 |
| $TiO_2$ | 76.53 | 80.31 | 0.78 | 0.00 | 0.00 | 0.00 | 0.00 | 0.00 | 0.00 | 0.00 |
| $Al_2O_3$ | 2.58 | 2.23 | 0.48 | 0.19 | 0.17 | 0.00 | 0.00 | 0.00 | 0.54 | 0.34 |
| FeO | 11.88 | 10.41 | 0.25 | 0.00 | 0.17 | 0.20 | 5.07 | 1.79 | 0.21 | 0.21 |
| MnO | 0.16 | 0.26 | 0.00 | 0.00 | 0.20 | 0.00 | 0.00 | 0.00 | 0.00 | 0.00 |
| $K_2O$ | 0.30 | 0.00 | 0.23 | 0.00 | 0.09 | 0.00 | 0.07 | 0.00 | 0.28 | 0.00 |
| $Na_2O$ | 0.00 | 0.00 | 0.00 | 0.00 | 0.00 | 0.00 | 0.00 | 0.00 | 0.00 | 0.18 |

The study of electron microscopic images of weathered limestones allows the following probable results of the impact of periodic cyclic freezing–thawing to be identified (Figure 4):

(1) The presence of extended fracture zones. In particular, the images with a high magnification (2.5–10 thousand times) show intermittent fractured areas where the continuity of the rock has been violated. Some fractured zones are impregnated with clayey matter (possibly due to the migration of clayey particles through channels in the rock). Finally, an important feature is the presence of intermittent microcracks of considerable length and variable width with separate bulges. For limestone samples, the most noticeable of these consist of an increase in the extent of fracturing, starting from the range of 46–69 microns (Table 3).

(2) Increased calcite microporosity. The high-resolution images clearly show the presence of numerous micropores 1–2 microns in size in calcite. Some large calcite grains have a pitted shape resulting from leaching.

(3) Deformation of the recrystallisation and metasomatic zones. Such zones are boreholes distinguished in SEM images by their large size in contrast to the major micro-grained part of the rock. These zones are rimmed by channels and cracks. The recrystallisation zones contain secondary calcite crystals up to 75 microns in size. Normally these zones are adjacent to the dolomitisation zones, although the development of these processes may not be simultaneous. The processes of leaching and corrosion of primary calcite crystals are very efficient.

(4) The presence of zones with carbonaceous matter, probably confined to large pores and cracks.

**Table 3.** Results of crack measurement and calculation in dolomitic limestone samples according to 3D X-ray tomography.

| Crack Size, μm | Borehole Core Samples | | Outcrop Surface Samples | |
|---|---|---|---|---|
| | Number of Fracture Crossings | Proportion of Cracks of a Particular Dimension Relative to Their Total Number, % | Number of Fracture Crossings | Proportion of Cracks of a Particular Dimension Relative to Their Total Number, % |
| <23.1 | 0 | 0.00 | 1 | 0.00 |
| 23–46 | 1 | 0.70 | 0 | 0.00 |
| 46–69 | 0 | 0.00 | 2597 | 75.60 |
| 69–92 | 62 | 43.10 | 454 | 13.20 |
| 92–116 | 28 | 19.40 | 353 | 10.30 |
| 116–139 | 26 | 18.10 | 25 | 0.70 |
| 139–162 | 19 | 13.20 | 4 | 0.10 |
| 162–185 | 7 | 4.90 | 0 | 0.00 |
| 185–208 | 1 | 0.70 | 0 | 0.00 |
| Sum | 144 | 100.00 | 3434 | 100.00 |

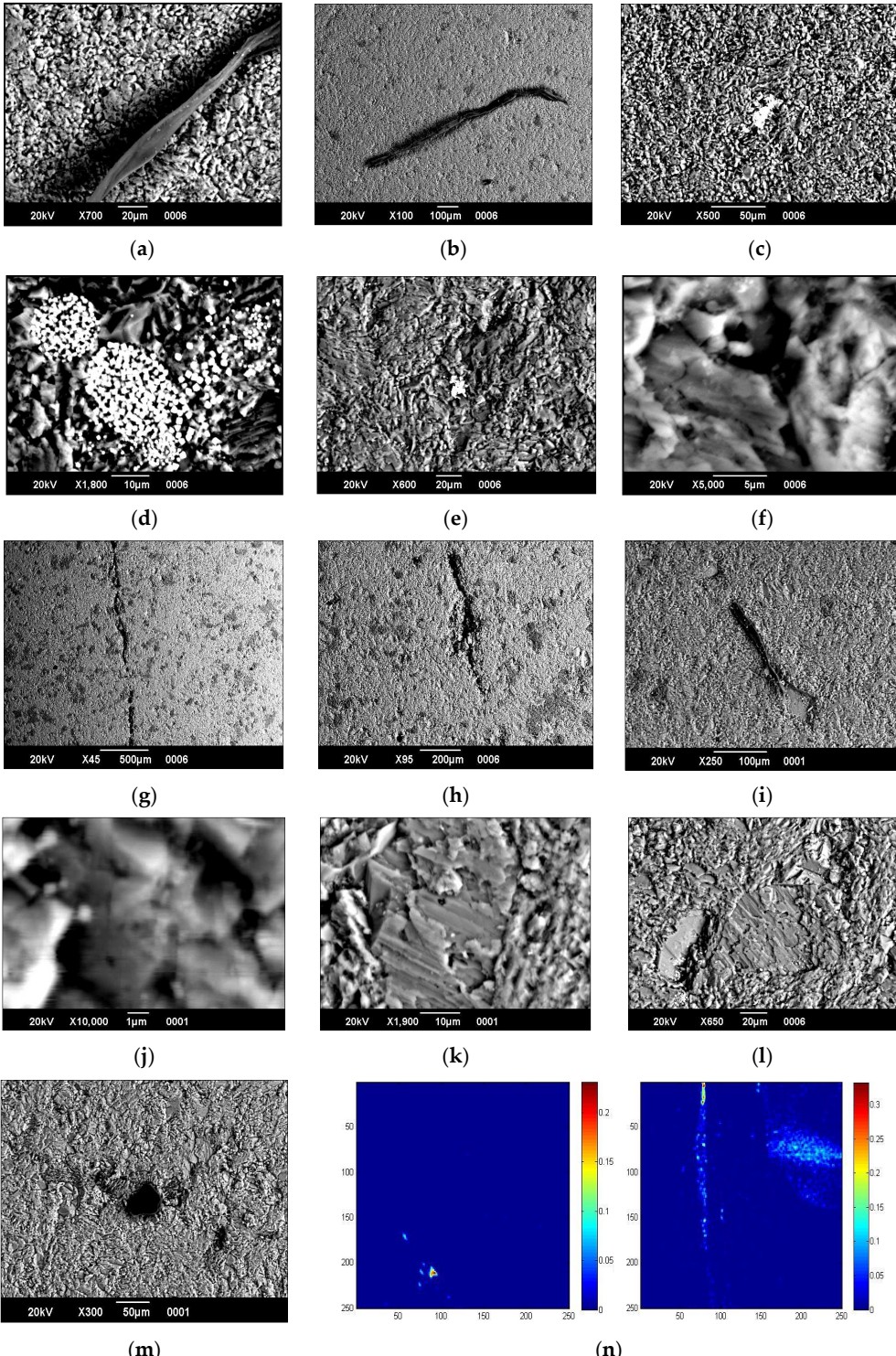

**Figure 4.** Scanning electron microscopic images of dolomitic limestone samples: (**a**) silicification zone in a "fresh" sample of limestone from a borehole; (**b**) the same, structural details; (**c**) baritisation zone (white in the centre) in a limestone sample from a borehole; (**d**) pyritisation zone (light) in a limestone sample from a borehole; (**e**) fractured intermittent areas with disrupted rock continuity in a weathered limestone sample; (**f**) the same, structural details; (**g,h**) microfracturing with individual bulges in the samples of weathered limestone; (**i,j**) the development of micropores 1–2 microns in size in the calcite of weathered limestones; (**k**) the "pitted" form of calcite grains in weathered limestone resulting from their leaching; (**l**) the deformation of recrystallisation and metasomatic zones in a weathered limestone sample; (**m**) carbonaceous matter (black), confined to pores and cavities in weathered samples; (**n**) fracture maps of "fresh" (**left**) and "weathered" (**right**) limestone sample.

### 3.2. Dolomitised Organogenic Limestone

According to electron microscopy data (Figure 5), organogenic dolomitised limestones sampled from boreholes were characterised by the irregular by volume spotty distribution of dolomitisation zones; uneven fracturing and porosity; the presence of silicification zones confined to dolomite areas; and the weak development of pyritisation and albitisation processes (along cracks). Weathered samples differed from borehole samples in terms of the intensive development of pyritisation, silicification and cavity formation processes. Pyrite formed separate large crystals up to around 0.5 mm in size, as well as filling the extended veinlet zones (probably replacing organic matter).

Caverns in calcite were generally filled with microgranular aggregates of spherical pyrite grains less than 1 μm in size. It is possible that dolomite may have formed over several stages, since large disseminations up to 200 microns in size were present along with microgranular aggregations. The processes of micropore formation in calcite were also developed.

The calcite was broken by microcracks into small blocks having uneven surfaces. The experimental results on cyclic freezing and thawing under laboratory conditions (1000 cycles) indicated the similarity of dolomitised organogenic limestone with the samples taken from outcrops. This was manifested, for example, in the character of structural calcite changes expressed by deformed, highly porous, fractured crystals with sculptured surface.

According to the chemical analysis data (electron microprobe analysis), "weathered" samples showed partial manganisation of dolomite zones, while manganese was absent in dolomite in the "fresh" samples. The presence of sulfatisation, phosphatisation and pyritisation zones, as well as clayey areas (with Al) (Tables 4 and 5), was also observed.

**Table 4.** Chemical composition of minerals of organogenic dolomitic limestone samples taken from the borehole core, wt. %.

| Oxide | Dolomite | Clayey Zone | Silification Zone | Calcite | Dolomite | Calcite | Feldspar Zone |
|---|---|---|---|---|---|---|---|
| CaO | 31.52 | 18.66 | 34.05 | 54.43 | 30.78 | 54.69 | 25.05 |
| MgO | 16.78 | 0.00 | 17.59 | 0.91 | 18.34 | 0.81 | 3.30 |
| SiO2 | 0.00 | 15.22 | 24.83 | 0.29 | 0.00 | 0.00 | 33.76 |
| $TiO_2$ | 1.12 | 62.50 | 0.74 | 0.00 | 0.23 | 0.00 | 11.74 |
| $Al_2O_3$ | 2.56 | 0.00 | 2.72 | 0.27 | 2.51 | 0.28 | 2.03 |
| FeO | 0.00 | 0.00 | 0.00 | 0.00 | 0.00 | 0.00 | 0.00 |
| MnO | 0.00 | 0.00 | 0.00 | 0.00 | 0.00 | 0.00 | 0.00 |
| $K_2O$ | 0.00 | 0.00 | 0.00 | 0.00 | 0.00 | 0.00 | 4.74 |
| $Na_2O$ | 31.52 | 18.66 | 34.05 | 54.43 | 30.78 | 54.69 | 25.05 |

**Table 5.** Chemical composition of minerals of organogenic dolomitic limestone samples taken from the surface of a rock outcrop, wt. %.

| Oxide | Silification and Baritisation Zone | Silification and Baritisation Zone | Silification and Baritisation Zone | Silification and Baritisation Zone | Dolomite | Calcite | Calcite |
|---|---|---|---|---|---|---|---|
| CaO | 45.89 | 53.90 | 45.56 | 13.14 | 32.49 | 54.75 | 32.83 |
| MgO | 1.61 | 0.57 | 4.70 | 2.02 | 16.61 | 1.03 | 16.63 |
| $SiO_2$ | 20.38 | 5.05 | 10.44 | 70.18 | 0.10 | 0.00 | 0.98 |
| $Al_2O_3$ | 1.69 | 0.36 | 2.56 | 1.58 | 0.00 | 0.00 | 0.00 |
| FeO | 1.45 | 0.12 | 3.39 | 1.46 | 2.50 | 0.00 | 2.05 |
| MnO | 0.12 | 0.00 | 0.13 | 0.00 | 0.08 | 0.00 | 0.00 |
| $K_2O$ | 0.85 | 0.00 | 1.08 | 0.62 | 0.00 | 0.00 | 0.00 |
| $Na_2O$ | 0.00 | 0.00 | 0.00 | 0.00 | 0.00 | 0.00 | 0.00 |
| $P_2O_5$ | 0.00 | 0.00 | 0.00 | 0.00 | 0.21 | 0.00 | 0.00 |
| $SO_3$ | 0.00 | 0.00 | 3.39 | 0.00 | 0.00 | 0.00 | 0.00 |
| BaO | 0.00 | 0.00 | 6.36 | 0.00 | 0.00 | 0.00 | 0.00 |
| SrO | 0.00 | 0.00 | 0.40 | 0.00 | 0.00 | 0.00 | 0.00 |

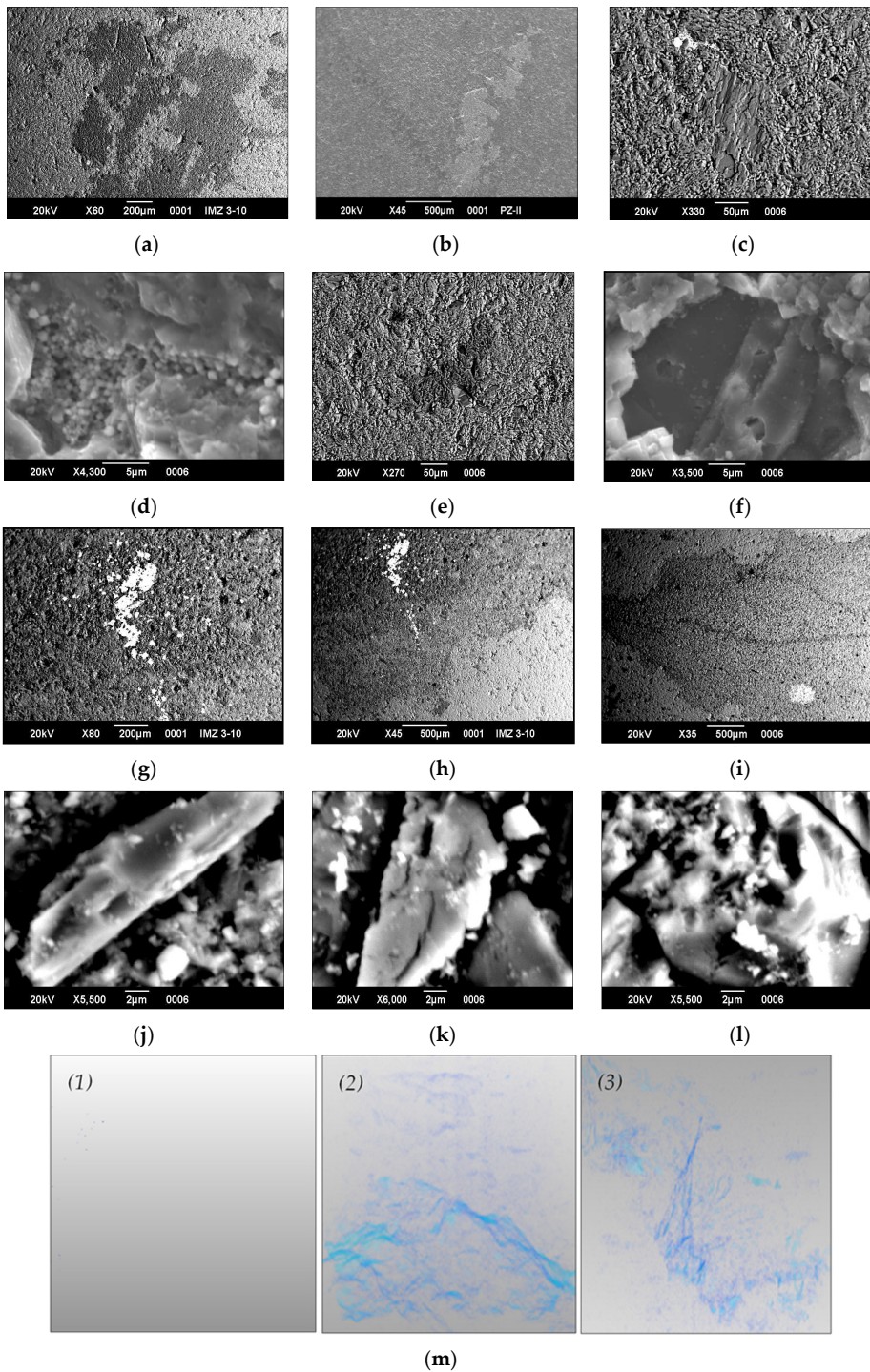

**Figure 5.** Scanning electron microscopic images and tomograms of organogenic limestone samples: (**a**) spotted distribution of dolomitisation zones (grey in the image) in "fresh" limestone samples from a borehole; (**b**) a sample of weathered limestone with a pyritisation zone (light); (**c**) sample of weathered limestone with pyrite in a cavity; (**d**) as previous, structural details; (**e**) sample of weathered limestone with micropores in calcite; (**f**) as previous, structural details; (**g**) silicified dolomite with pyrite in a weathered limestone sample; (**h**) junction of the silicification zone and calcite (light) in a weathered limestone sample; (**i**) development of dolomitisation zones (gray) in weathered limestone sample; (**j**) the deformation of a calcite crystal in a sample taken from mime outcrop; (**k,l**) deformation of a calcite crystal in limestone samples subjected to 1000 cycles of freezing and thawing under laboratory conditions; (**m**) sections of tomograms of organogenic limestone samples (sample size 4.0 cm × 4.0 cm × 4.0 cm): (1)—tomogram of the "fresh" sample, (2)—tomogram of the same sample after 1000 cycles of freezing and thawing under laboratory conditions, (3)—tomogram of a weathered sample taken from the outcrop surface.

Thus, areas with organogenic substances and calcite–dolomite junctions were shown to be subject to intense effects of weathering processes.

The determination of the strength properties of limestones turned out to be quite informative in terms of establishing changes in their structural bonds during cryogenesis. The limestone samples taken from the borehole had a uniaxial compressive strength of 33–46 MPa and a density of 2.66–2.71 g/cm$^3$. At the same time, the strength of samples taken from the walls of the quarry was in the range of 24–36 MPa; here, the density decreased to 2.53–2.63 g/cm$^3$. A decrease in physical and mechanical properties was also noted during repeated freezing and thawing of soils (1000 cycles) in the laboratory; here, the ultimate strength of samples taken from boreholes decreased to values of 30–40 MPa. Moreover, for some samples, when subjected to more than 700 thermal shock cycles, a violation of their integrity was observed (Figures 6 and 7).

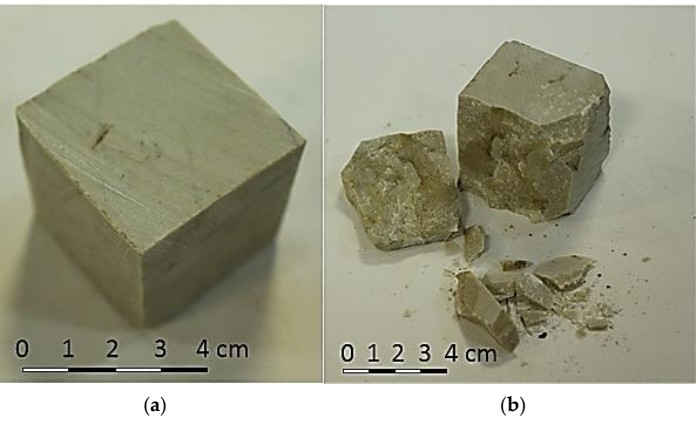

(**a**)　　　　　　　　　　　　　　　(**b**)

**Figure 6.** Disintegration of dolomitic limestone sample during repeated freezing and thawing: (**a**) soil sample prepared from borehole cores; (**b**) disintegration of the soil sample after 780 cycles of alternating freezing and thawing under laboratory conditions.

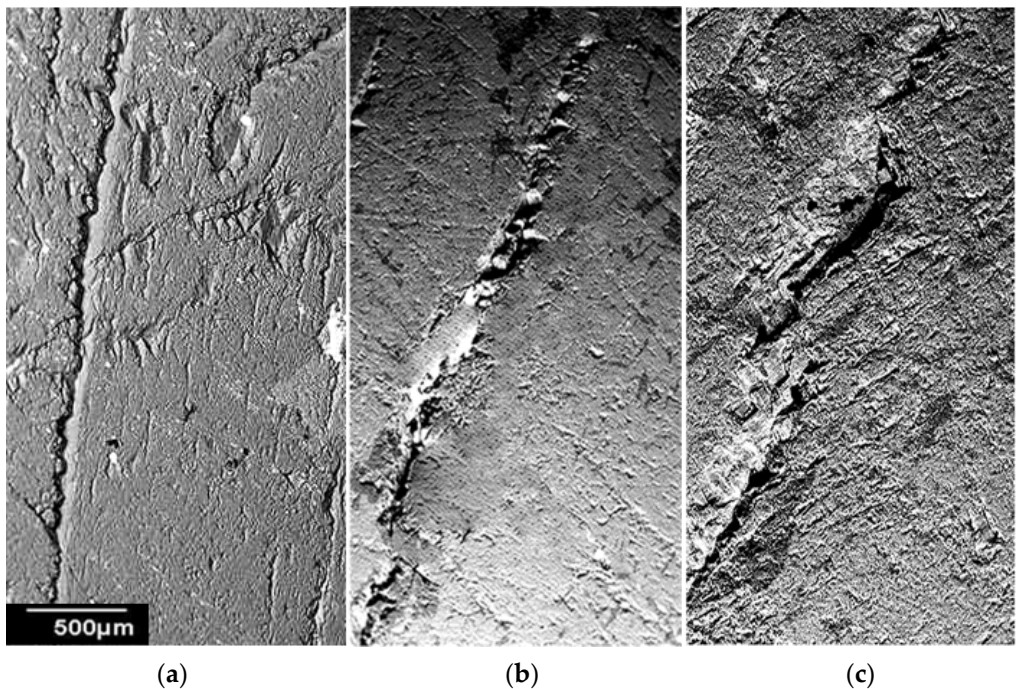

(**a**)　　　　　　　　　　　　(**b**)　　　　　　　　　　　　(**c**)

**Figure 7.** Growth of a crack in a dolomitic limestone sample during repeated freezing and thawing under laboratory conditions: (**a**) crack in the soil sample prior to laboratory testing; (**b**) crack after 520 cycles of alternate freezing and thawing of the sample; (**c**) crack after 990 cycles of alternate freezing and thawing of the sample.

## 4. Discussion

Research has shown that recrystallisation and metasomatism are the main natural processes of trans-formation of the mineral composition of primary limestone. These processes lead to the appearance of areas with larger secondary calcite crystals and metasomatic minerals such as dolomite, barite, pyrite, etc. Such areas are less resistant to external influences—in particular to cryogenesis—due to the formation of voids between calcite crystals, channels and other defects, which become available sites for the development of ice crystals in them. The growth of the latter causes a break in the continuity of the rock. That is, the process of formation of recrystallized and metasomatically transformed calcite itself is a certain effect on the primary rock. The result of this action is a certain deformation of the structure, due to which the areas of more intense recrystallization become less resistant to various other external influences.

According to the structural analysis, the sufficiently high resistance dolomitic limestones to processes of cyclic freezing and thawing might be assumed. At the same time, possible areas and zones in which these processes can have a significant impact are characterised by increased clay content, as well as pyritisation, dolomitisation and baritisation.

The limestone moistening regime plays a significant in the intensity of cryogenesis. This may explain the fact that, in comparison with samples taken from the surface of the mountain outcrop, the strength indicators of limestone samples obtained under laboratory conditions are somewhat lower for the same time period of "weathering". The modelling of "weathering" carried out under laboratory conditions took place without additional or periodic moistening. Evidently, the change in the chemical composition of primary limestones can be explained in terms of the intense movement of aqueous solutions in the upper parts of the studied carbonate massif, along with the appearance in their composition of phosphates, sulfates, chlorides, manganese and alumina.

Thus, in order to predict changes in the properties of soils during cryogenesis, it is important to solve the issue of developing a methodology for the laboratory testing of soils. The technique should take into account many factors that determine the intensity of weathering of soil massifs. With regard to South Yakutia, very little purposeful identification of such factors and their features has been carried out. Those individual works that have been carried out are fragmented and fail to take into account the complexity of the subject.

## 5. Conclusions

The comparison of investigated varieties of carbonate soils sampled from mine surfaces with "fresh" samples subjected to periodic cyclic freezing and thawing under laboratory conditions demonstrates the similarity of the changes occurring in them. However, the severity and intensity of transformations in soils subjected to temperature effects under laboratory conditions are significantly lower. It is evident that soils are influenced by a wider range of environmental factors under natural conditions. Since previous works studying the specific conditions in South Yakutia are rather fragmentary and limited to short-term observations, it was necessary to identify such factors and their features. Thus, the significant influence of alternating temperatures on changes in the composition and structure of South Yakutian soils have been comprehensively described.

**Author Contributions:** Conceptualization, A.M.; Funding acquisition, A.M. and Z.Z.; Investigation, A.K. and V.R.; Methodology, A.M. and A.K.; Writing—original draft, A.M.; Writing—review and editing, A.K. and A.M. All authors have read and agreed to the published version of the manuscript.

**Funding:** The presented study was funded by the RFBR and NSFC in accordance with research project No. 20-55-53006 and No. 42011530083, and supported by the National Natural Sciences Foundation of China in accordance with research project No. 41771078.

**Institutional Review Board Statement:** Not applicable.

**Informed Consent Statement:** Not applicable.

**Data Availability Statement:** The data presented in this study are available on request from the corresponding author.

**Conflicts of Interest:** The authors declare no conflict of interest. The funders had no role in the design of the study; in the collection, analyses or interpretation of data; in the writing of the manuscript; or in the decision to publish the results.

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
