# Peer review of "Carbonate Soil Cryogenesis in South Yakutia (Russia)"

_minerals, doi:10.3390/min11080800_

Round 1

Reviewer 1 Report

Dear Authors,

The topic of your manuscript is very interesting, the text is very well strucured and very good understandable.

Some comments for improvement:

  • Line 14, 16, 40: as... as.. (as borehole as zone) - improve expression
  • Introduction:
    • add a short definition of soil cryogenesis (Soil cryogenesis describes...)
    • Rock vs. rocky soil vs soil: explain/define shortly in more detail
  • Materials and Methods:
    • Add a short geological information of the investigation area
    • Add a short climatic description of the are (time period of winter, temperatures, etc.)
    • Give more detailed information about sampling (depth of rock sampling etc.)
    • Information about the rock strength of the samples would be nice if available (indication for degree of weathering and structure defects)
    • More detailed information about freezing cycles + show temperature curve (are the experiments corrispondent to natural conditions?)
    • Fig. 1: add a picture of the sample itself (picture of handpiece)
  • Chapter 3.1 / 3.2: numbering of heading
  • Table 3: some units are missing, define "fracture fraction"
  • Tables (especially Tab. 3, 4, 5): describe content in more detail in text
  • Fig. 3 bottom line: not very conclusive (scale?, etc.), may add a picture of the samples (handpiece) itself
  • Discussion: don´t start with "Thus", mybe "The investigations have shown..
  • Pictures of investigated material and/or sampling location must be added somewhere in the beginning

Many thanks and best regards

Author Response

Dear Reviewer! In the process of responding to all your comments, we hope that our work has been improved. Changes made to the work are as follows:

1) In addition to the qualitative research results, the abstract includes quantitative indicators.

2) A description of the natural and climatic conditions pertaining to the study area (geological, geocryological, climatic) has been provided.

3) Detailed information on the method of soil sampling for testing and conducting experiments on cyclic freezing and thawing of samples, explanatory diagrams and photographs has been adduced.

4) the strength and density characteristics of the studied soils have been provided.

5) additional explanations to the figures and explanations for the terms used in the text of the article have been introduced.

6) The text has been translated and proofread by a native English speaker.

Reviewer 2 Report

The study investigated the use of 1000 freezing/thawing cycles to simulate natural weathering of carbonate rock samples by comparing mineral compositions of newly collected and 10-year old borehole samples. This kind of study is interesting and fit well with the journal scope. But there is a lack of sufficient background and references in the introduction and no in-depth discussions. The English and grammar is poor and should be carefully proofread

L 14, 16,40, 56, 107…….: “as borehole as…”, is this right?

L28: what do you mean by “Text Soils”?

Abstract: mostly qualitative description, please add quantitate descriptions

Introduction, need more background and references

L61-80: replace the bullets with numbers or letters

L85-99: these are not research content, please remove

There is a lack of in-depth discussions

Author Response

Dear Reviewer! In the process of responding to all your comments, we hope that our work has been improved. Changes made to the work are as follows:

1) In addition to the qualitative research results, the abstract includes quantitative indicators.

2) A description of the natural and climatic conditions pertaining to the study area (geological, geocryological, climatic) has been provided.

3) Detailed information on the method of soil sampling for testing and conducting experiments on cyclic freezing and thawing of samples, explanatory diagrams and photographs has been adduced.

4) the strength and density characteristics of the studied soils have been provided.

5) additional explanations to the figures and explanations for the terms used in the text of the article have been introduced.

6) The text has been translated and proofread by a native English speaker.

7) The authors suggest that the expansion of the "Introduction" section may distract the reader from the main idea of the article. In addition, there are very few literary sources on the cryogenesis of carbonate soils in the studied region (units). The analytical work performed, as well as the methodological approach applied by the authors in the study of the carbonate strata of the region can be attributed to the pioneer ones.

Round 2

Reviewer 2 Report

The revised version has been significantly improved.

some minor comments: Table 1 thorugh 5, use 2 digits for all values to keep consistency

Author Response

Dear reviewer, thank you for your comments! The tables have been edited. Numbers in tables have the same dimension - two decimal places.